# Management Strategies for Antidepressant-Related Sexual Dysfunction: A Clinical Approach

**DOI:** 10.3390/jcm8101640

**Published:** 2019-10-07

**Authors:** Angel L. Montejo, Nieves Prieto, Rubén de Alarcón, Nerea Casado-Espada, Javier de la Iglesia, Laura Montejo

**Affiliations:** 1Psychiatry Service, Clinical Hospital of the University of Salamanca, Institute of Biomedical Research of Salamanca (IBSAL), Paseo San Vicente SN, 37007 Salamanca, Spain; nievesprieto2010@gmail.com (N.P.); ruperghost@gmail.com (R.d.A.); nmcasado91@gmail.com (N.C.-E.); javidelaiglesia.jdli@gmail.com (J.d.l.I.); 2University of Salamanca, EUEF, Donantes de Sangre Street S/N, 37007 Salamanca, Spain; 3Barcelona Bipolar and Depressive Disorders Program, Institute of Neurosciences, University of Barcelona, IDIBAPS, CIBERSAM, Hospital Clinic of Barcelona, 08401 Catalonia, Spain; laumonteg@gmail.com

**Keywords:** antidepressant, sexual dysfunction, erectile dysfunction, anorgasmia, orgasm retardation, TESD, management strategies, treatment

## Abstract

Major depressive disorder is a serious mental disorder in which treatment with antidepressant medication is often associated with sexual dysfunction (SD). Given its intimate nature, treatment emergent sexual dysfunction (TESD) has a low rate of spontaneous reports by patients, and this side effect therefore remains underestimated in clinical practice and in technical data sheets for antidepressants. Moreover, the issue of TESD is rarely routinely approached by clinicians in daily praxis. TESD is a determinant for tolerability, since this dysfunction often leads to a state of patient distress (or the distress of their partner) in the sexually active population, which is one of the most frequent reasons for lack of adherence and treatment drop-outs in antidepressant use. There is a delicate balance between prescribing an effective drug that improves depressive symptomatology and also has a minimum impact on sexuality. In this paper, we detail some management strategies for TESD from a clinical perspective, ranging from prevention (carefully choosing an antidepressant with a low rate of TESD) to possible pharmacological interventions aimed at improving patients’ tolerability when TESD is present. The suggested recommendations include the following: for low sexual desire, switching to a non-serotoninergic drug, lowering the dose, or associating bupropion or aripiprazole; for unwanted orgasm delayal or anorgasmia, dose reduction, “weekend holiday”, or switching to a non-serotoninergic drug or fluvoxamine; for erectile dysfunction, switching to a non-serotoninergic drug or the addition of an antidote such as phosphodiesterase 5 inhibitors (PD5-I); and for lubrication difficulties, switching to a non-serotoninergic drug, dose reduction, or using vaginal lubricants. A psychoeducational and psychotherapeutic approach should always be considered in cases with poorly tolerated sexual dysfunction.

## 1. Introduction

Treatment emergent sexual dysfunction (TESD) is common in the sexually active population. TESD is one of the most long-lasting side effects in users of antidepressants (ADs), despite its occurrence being underestimated in technical data sheets for ADs, where the incidence is stated as 2%–16% [1]. These percentages are remarkably low compared with those that are estimated in everyday praxis and specific studies [2,3,4] including systematic reviews [5,6] and meta-analysis [7] (Table 1). The reasons for this underestimation include the fact that the incidence of sexual dysfunction (SD) is obtained from registered drug trials and short-term efficacy studies, which are unreliable since they either include sexually inactive patients [5] or do not use specific and validated questionnaires, rather relying solely on spontaneous reports. Using large samples of patients, the true prevalence of TESD has been calculated in everyday clinical praxis [8,9] to be significantly greater than the prevalence stated in technical data sheets for ADs. TESD is believed to be twice as common in depressed patients than in controls [10]. Since sexual dysfunction (SD) is widely associated with depression, screening for TESD as a possible depressive symptom [11] is advisable when screening for affective disorders in people with TESD [12].

Specific questionnaires are available to determine the presence of TESD. When employing some of these in sexually active patients, such as the Psychotropic-Related Sexual Dysfunction Questionnaire (PRSexDQ; also known as SALSEX) [13] or the Changes in Sexual Function Questionnaire (CSFQ) [14], the reported prevalence of TESD was 50%–70%, and was higher in patients being treated with selective serotoninergic reuptake inhibitors (SSRIs) [15,16,17,18]. Furthermore, the prevalence of TESD was found to be higher than 80% in healthy volunteers that took an SSRI for at least eight weeks [19,20]. The consequences of TESD are various, ranging from a decline in the quality of life to treatment drop-out for the same reason [18,21] (however, the exact percentages of patients who do this are unclear [22,23]). This poses a potential risk of relapse and other negative repercussions regarding depression prognosis [24,25]. The most frequent symptoms of TESD are decreased sexual desire and orgasm retardation [2,18]. Erectile dysfunction is less frequent (although the prevalence is 30%–40% for the ADs paroxetine, citalopram, and venlafaxine at the usual therapeutic dosing) [18]. Anorgasmia or lack of ejaculation are the side effects that are least well tolerated by patients, especially males [19], who tend to report it more [26].

Additionally, TESD often remains unaddressed in everyday praxis unless the clinician directly asks the patient, which has been found to be uncommon in primary care [27,28,29]; this causes the prevalence of TESD to be underestimated in depressed patients [30,31]. Spontaneous communication from patients about this side effect is not above 15%–20% [18]. TESD usually causes distress for the patient and/or their partners in the short-, medium-, and long term. This aspect of sexual tolerability should be carefully considered in the selection of an AD [32].

SSRIs, as well as venlafaxine and clomipramine, are the ADs that are most commonly associated with TESD, while other ADs with a different, non-serotoninergic mechanism of action (mirtazapine, bupropion, agomelatine, and moclobemide) seem to be associated with a lower prevalence of TESD [16]; controlled studies and case series of these drugs in the medium- and long term showed little decline in patients’ sexuality compared with that observed for serotoninergic ADs [33,34,35,36]. However, the antidepressive effect of non-serotoninergic ADs can be insufficient in some groups of patients (those with obsessive-compulsive disorder, panic disorder, or anxiety disorders) who require a serotoninergic effect.

In general, there is a scarcity of well-designed comparative studies of contemporary ADs involving a direct assessment of sexual side-effects as the primary outcome measure [37]. Dual ADs, i.e., mixed reuptake inhibitors of both serotonin and noradrenaline (duloxetine, desvenlafaxine), could be associated with a lower incidence of TESD, however contradictory data have been reported in diverse studies with different designs [38].

The effects of duloxetine on the sexual lives of patients seems to be linked to its antidepressive effects, with SD being more frequent in populations that present a lesser response to the drug than in those that respond adequately [39]. On the other hand, vortioxetine, which has a multimodal mechanism of action which is serotoninergic, noradrenergic, and dopaminergic, seems to be associated with a lower prevalence of TESD, in agreement with data derived from registered clinical trials [40,41] and from a specific study that switched patients’ AD to vortioxetine when TESD appeared [42]; however, there is not enough naturalistic data from everyday clinical praxis that confirms this observation.

Desvenlafaxine has been shown to be associated with a lower incidence of SD side effects (e.g., erectile dysfunction or anorgasmia) than other ADs [43,44]; however, these studies were not specifically designed to detect alterations in sexual function. One study that was specifically designed to detect alterations in sexual function, a post-hoc analysis of a drug trial that was compared with a placebo using the Arizona Sexual Experiences Scale (ASEX), found that, when compared to placebo, two groups of females had a similar incidence of orgasm retardation [45], while other sexual functions such as sexual desire and arousal were not affected. The results of another study suggest that the incidence of TESD could be higher in patients taking 100 mg of desvenlafaxine than those taking 50 mg [46]; however, these results are not conclusive since no statistically significant differences were observed.

Given the wide array of AD drugs that are available and their potential for TESD, in the present study, we aimed to gather information about possible management strategies for TESD in patients who are taking an AD which does not compromise the antidepressive effect. We aimed to gather and update relevant information, including studies conducted with relatively new ADs (desvenlafaxine, vortioxetine).

## 2. Methodology

We performed a review of TESD and its possible management strategies. There was no specific time-delimitation. Priority was given to literature reviews and articles that had been published most recently, with special attention given to articles that had been subjected to peer review. PubMed and Cochrane were the main databases used. The following search terms and their derivatives were used in multiple combinations: “sexual dysfunction”, “erectile dysfunction”, “sexual function*” (to allow for both “function” and “functioning”), “orgasm retardation”, “anorgasmia”, “antidepressant”, and “antidepressive”.

We excluded articles which focused on SD as a general entity rather than as a consequence of treatment with ADs. Initially, 124 relevant results were found which were focused on TESD. Upon manual screening and categorizing, these results included (1) those which acknowledged the prevalence and clinical relevance of the phenomenon, and (2) those which focused on clinical management strategies for TESD.

The results described under “(1)” were mainly used for Section 1 of this paper. There were a total of 32 articles, including 4 transversal and cross-sectional studies, 11 observational studies, and a total of 17 reviews (of which 5 were meta-analyses). Several of these articles were also used on other sections of the paper to provide background, since many of them are antidepressant-specific.

The results described under “(2)” are included in Section 3 of this paper. There were a total of 72 articles, including 29 reviews (of which 7 were systematic reviews or meta-analyses), 19 experimental studies, and 23 observational studies. An additional 12 articles were included after selected areas of the paper were revised (experimental studies with sildenafil, trazodone studies, sex surrogate studies, and validated questionnaires about SD). The Zotero reference management tool was used to build a database of all the considered articles. 

## 3. Results and Clinical Approach

The most practical way to approach TESD seems to be following a simple three-step sequence: (1)Prevent TESD in susceptible population;(2)Conduct routine checks for TESD in sexually active patients who are prescribed ADs;(3)Perform clinical intervention when TESD is a problem for the patient and/or their partner or causes a potential risk for treatment drop-out.

### 3.1. Primary Prevention: Using ADs with a Low Incidence of TESD

TESD frequently appears in sexually active patients and should be prevented as much as possible. Before deciding on an AD treatment—which could greatly affect the patient’s sexual life, in some cases for a prolonged period of time—all interviews with patients should include questions about their sexual life, frequency of intimate relations, sexual satisfaction, and the relevance of these aspects for the patient (and their partner, if they have one). In most patients, the initial level of concern about TESD is low at the beginning of treatment or in short-term treatments, however some patients will abandon long-term treatment (beyond 6–12 months) if TESD persists [47]. However, other patients, such as post-menopausal women, accept TESD with little trouble and do not consider it a major problem in their lives [18]. In patients (with or without partners) who consider preserving sexual function as relevant, the ethical obligation of discussing possible sexual side effects of AD medication should always be kept in mind before prescribing it so that the patient can make an adequately informed decision about their treatment. Psychotherapy may be the first choice for patients who do not want to receive ADs due to the risk of SD.

As a first option, clinicians should consider commencing treatment with ADs that have a low incidence of TESD, namely agomelatine, mirtazapine [3,5,48], or bupropion [49,50]. Bupropion is the AD for which there is the most scientific evidence for a low incidence of TESD and could therefore be the first option in the sexually active population. While other non-serotoninergic drugs are also associated with a low incidence of TESD, such as mianserin [51,52,53], maprotiline, and reboxetine [54], these have other adverse effects that need to be carefully considered, such as possible weight gain, sedation, and medium- to long-term anticholinergic effects. 

If an AD is required to have some serotoninergic effect and also to have a lower impact on sexual function and orgasm/ejaculation, fluvoxamine is the best candidate when not dosing above 100 mg/day [2]. Among drugs which are partially serotoninergic, the possibilities are desvenlafaxine (when dosing under 100 mg/day) [46] and vortioxetine (10–15 mg per day) [55]. A prospective naturalistic study of desvenlafaxine conducted by our Sexuality and Mental Health project group [56] showed promising results when prescribing this drug as the first AD to a group of depressed patients, with an incidence of TESD of 44.4% being observed. This incidence is lower than that observed in a previous study that used the same methodology for SSRIs and serotonin–norepinephrine reuptake inhibitors (SNRIs) (66% for SSRIs and 75.4%–74.8% for SNRIs) [16]. Moreover, in two international studies of the prevalence of TESD, a large proportion of the participating patients showed new TESD symptoms or the worsening of pre-existing TESD symptoms during the first weeks after beginning treatment with SNRIs (27%–65% of females and 26%–57% of males) [4,57].

It is necessary to adequately clarify the role that different ADs play in the development of TESD and the subsequent decline of patients’ quality of life. In patients who are sexually active, prescribing drugs that do not worsen patients’ sexual lives should always be kept in mind when treating depressed patients. Furthermore, since TESD is most frequent for some concrete ADs, it should always be considered when researching new molecules and searching for better profiles of medium- and long-term tolerability [32,58].

### 3.2. Detection and Exploration of TESD

Once an AD treatment is prescribed, especially if a serotoninergic AD is chosen, it is essential to discover whether TESD exists by routinely asking clear and simple questions such as “How has your sexual life been since you started taking the medication?” and “Have you noticed any change that worries you?”. A small proportion of patients may be surprised by such questions and will not complain until specifically and repeatedly asked about every sexual aspect one after another. Therefore, if the answer to the first question is “No”, it is best to make sure by asking the question “Have you noticed any problems in your sexual desire, your orgasms/ejaculation, or your arousal capacity since you started taking the medication?”

Openly discussing sexuality with patients might be difficult for some clinicians [59,60,61] for several reasons (uncertainty, fear of causing distress, etc.). Clinicians should always approach this issue in a neutral manner, expressing concern about the patient’s wellbeing and informing them about the potential sexual side effects of AD medication. An appropriate sexual medical history must be used to thoroughly understand the onset of the first SD symptoms, SD severity, the impact of SD on the sexual lives of the patient and their partner, and the influence of SD on the patient’s quality of life.

The validated PRSexDQ-SALSEX is an adequate option to better quantify TESD and measure its degree of acceptance by the patient. The questionnaire includes a brief (no more than three minutes) global exploration through five items and can be self-applied or applied through direct interviewing. The maximum score is 15 points, and scores are subdivided into the following grades of dysfunction: low (0–5 points), moderate (6–10 points), and severe (11–15 points). The last item indicates how relevant the TESD (if present) is to the patient, whether it is affecting their quality of life, and whether it induced them to drop out of the treatment. The PRSexDQ-SALSEX questionnaire can be freely downloaded for scientific purposes and by researchers at [62].

Other well-validated questionnaires that are used to measure TESD include the CSFQ and ASEX [63], the latter of which can be accessed at [64] and the Sex Effects Scale (SexFX) [65]. 

To further measure the severity of SD, potentially valid options for males include the International Index of Erectile Function (IIEF) and the Sexual Encounter Profile (SEP) [66], while potentially valid options for females include the Sexual Interest and Desire Inventory-Female (SIDI-F) [67]. It is necessary to consider patient expectations regarding the assessment of sexual function by their prescriber. Provider discomfort and uncertainty about discussing sexual functioning with their patients may be a barrier. Unfortunately, there are no studies regarding this important consideration in the literature, although it is certainly of great relevance for the early identification and management of TESD.

Once TESD is identified, if it is poorly tolerated by the patient (or their partner) or negatively impacts their quality of life, it may be possible to intervene depending on the specific problem(s) posed by, and individual clinical symptoms of, TESD. In some occasions, it will not be possible to fully restore previous sexual function. However, at least partially alleviating the experienced symptoms can satisfy the patient’s expectations without jeopardizing the antidepressive effect of their treatment. 

### 3.3. Intervention

If TESD cannot be avoided, it is necessary to intervene with the aim of eliminating it at least partially, if not completely. Research on the elimination of TESD is scarce and there is a dearth of robust scientific evidence, with studies being mostly non-controlled naturalistic studies. Generally, the most widely used and empirically contrasted methods for treating TESD in everyday praxis (which have their own risks and benefits) are the following: (1) waiting for spontaneous remission; (2) dose reduction; (3) the addition of an antidote such as phosphodiesterase 5 inhibitors (PD5-I) if there is erectile dysfunction; (4) withdrawal from the AD for 24–48 h before sexual relations in the case of anorgasmia; (5) switching to another non-serotoninergic AD [6,68,69,70]; and (6) non-pharmacological measures. However, none of these methods is devoid of risks—some may carry the risk of relapse or the appearance of new concomitant side effects related to the new treatment—and clinicians should therefore offer their patients the treatment option that best suits them. 

#### 3.3.1. Waiting for Spontaneous Remission

Waiting for spontaneous remission is the most widely used method by physicians to treat TESD, however it is also the least effective [2,71,72]. Some patients experience partial or total improvement of their TESD within a few weeks or months [73], while others only experience it for a few days before it rapidly disappears. However, most patients (80%) do not present any improvement after six months of treatment (Table 2).

#### 3.3.2. Lowering the Dose of or Withdrawing AD Treatment

There is a great individual variability in TESD depending on pharmacodynamic and genetic aspects, with some patients experiencing TESD at low dosages and others experiencing no TESD at high dosages [2,39]. Halving the dose of an AD improves TESD in 75% of cases, with total recovery being almost guaranteed after a few days or weeks of withdrawal from the medication [73].

Patients should be carefully selected for withdrawal from ADs in order to avoid relapses due to premature withdrawal, and psychotherapy should always be considered as an alternative or an add-on treatment. On some occasions, reducing the dosage of a serotoninergic drug may improve TESD but worsen the depressive symptoms [2], increase anxiety levels, or cause the onset of discontinuation symptoms that should be adequately monitored. Reducing doses gradually can help to prevent withdrawal syndrome.

In such cases, dosage reduction in association with another non-serotoninergic ADs (e.g., agomelatine, bupropion, or mirtazapine) could potentially be useful [3,5,48,49,50], however there is a lack of controlled studies to support this strategy.

#### 3.3.3. Potentiation Strategy: Addition of an “Antidote” or Add-On Treatment

A great number of substances have been described for the treatment of TESD, although none of them are devoid of inconveniences [74,75,76] (Table 3). Occasionally, these drugs can be administered a few hours before sexual relations or as a permanent measure. They act through different mechanisms, namely dopaminergic agonists [77] (amantadine [78], dextroamphetamine and pemoline, ropinirole), serotoninergic antagonists (cyproheptadine) [79,80], 5HT1A receptor stimulants (buspirone [81]), adrenergic agonists (yohimbine [82], pentoxifylline [83], fluparoxan), cholinergic agonists (neostigmine, bethanechol) [84], antioxidant pathways (pycnogenol [85]), or through poorly understood mechanisms (*Gingko biloba* extract [86]).

However, these methods present clear inconveniences, with very dissimilar results regarding efficacy, and some can even cause other, less well-tolerated, adverse effects. Furthermore, there is scarce evidence to support the use of these alternatives, since studies have consisted of short case series or anecdotal case reports with conflicting results. The addition of bupropion seems to be an exception, with robust scientific evidence supporting its clinical usefulness [50]—namely, three randomized, double-blind, placebo-controlled studies in which a potentiation strategy with bupropion was found to improve sexual function. However, clinicians should be aware that adding bupropion may worsen anxiety levels in some patients. 

The addition of 5HT2 blocker ADs (mianserin [51,52,53] and mirtazapine [87]) can have good results in treating TESD, however the weight gain that may result can be poorly tolerated, especially among women.

In cases of erectile dysfunction, PD-5 inhibitors such as sildenafil, vardenafil, and tadalafil have been proven to be effective versus placebo for treating erectile dysfunction secondary to psychoactive drugs (69%) and are well tolerated [88,89,90]. Additionally, these inhibitors may have an antidepressive effect by blocking central cholinergic receptors; recent experimental studies with mice have proved that administration of sildenafil potentiates antidepressant-like activity of sertraline, maprotiline, trazodone [91], bupropion, venlafaxine [92], mianserin, tianeptine [93], and amitriptyline [94]. However, sildenafil’s mechanism of action may be directly opposed to the one used by paroxetine [95], and when used together the antidepressant effect may prove diminished or nullified. Furthermore, PD-5 inhibitors do not improve low sexual desire or retardation in orgasm/ejaculation. Another disadvantage of these medications is that they require previous planning of sexual activity; however, tadalafil has a half-life of more than 36 h, which could possibly remove this requirement.

Additionally, pycnogenol has shown promising results when used as an add-on to escitalopram; it was found to attenuate TESD when it was used in the first month of treatment and treatment was continued for two consecutive months [85]. This may be due to its ability to improve endothelial functions through its antioxidant, anti-inflammatory, vasodilatory, and anticoagulant action. However, elevated heart rate was observed as a side effect, and precaution is therefore recommended in patients with cardiovascular pathology.

The addition of aripiprazole, due to its partially agonistic dopaminergic effect and its 5HT2 receptor antagonism, has proven to be effective in improving sexual desire and sexual satisfaction in monotherapy refractory depression, although only in women [96]. The addition of bupropion or testosterone gel [97] has also been shown to be beneficial in treating TESD.

#### 3.3.4. Short-Term Interruption Periods or “Weekend Holidays”

Short-term interruption periods, or “weekend holidays”, consist of withdrawing the AD 48–72 h before having sexual relations and reintroducing it afterwards [98]. This strategy can be useful in patients for whom treatment cannot be interrupted and whose primary SD is anorgasmia and could be used for patients who are prescribed SSRIs (except fluoxetine, due to its prolonged half-life of >14 days). Although this method is usually well tolerated by patients, it may cause some negative effects, such as lack of adherence and the presentation of discontinuation syndrome (which involves dizziness, nausea, vertigo, insomnia, and anxiety) after the sudden withdrawal from the drug. Patients need to be informed about these risks.

A possible alternative could be lowering the dose to a half for two consecutive days a week [69] prior to having sexual relations in that 24–48 hour time window, before subsequently continuing with the usual dose. In theory, temporarily reducing plasma levels of the AD, could improve erection/lubrication quality and orgasm alterations; however, it could not improve the lack of sexual desire. 

The usefulness of this strategy is mostly supported by a non-controlled study with a small sample size (30 patients); more recent, methodologically sound, controlled trials of this strategy are currently lacking [99].

#### 3.3.5. Switching to Another AD Medication

Most studies and therapeutic alternatives for TESD focus on switching to another AD medication which does not produce the side effect of TESD, has a different mechanism of action, and is both effective and well tolerated by the patient [99].

##### Switching to A Non-Serotoninergic Antidepressant

Patients with TESD caused by an SSRI are not likely to experience improvement in their TESD when switching to another SSRI due to the similar mechanism of action. Accordingly, one possible way to alleviate TESD is to switch to a non-serotoninergic or partially serotoninergic drug. This can be a useful approach for all TESD-related symptoms as long as precautions are taken to avoid discontinuation syndrome.

Switching to fluvoxamine from another SSRI could improve anorgasmia/ejaculation [2] due to the low incidence of anorgasmia in fluvoxamine, although there are no specific studies supporting this hypothesis. Switching to other non-serotoninergic drugs (agomelatine, bupropion, mirtazapine, etc.) is useful in the short- and medium term, although the therapeutic response to the switch needs to be carefully monitored [3,5,48,49,50]. To avoid discontinuation syndrome, it is advised to slowly lower the serotoninergic dosage (sometimes for several weeks) while simultaneously gradually increasing the dosage of the new AD. Another alternative could be to substitute an SSRI for fluoxetine for about 7–14 days, since its longer half-life could prevent discontinuation symptoms.

##### Switching to Bupropion

Bupropion is the most widely used AD due to its positive performance in numerous trials in the United States [100]. It leads to the release of dopamine and noradrenaline without affecting serotonin. Controlled studies have shown good results when switching fluoxetine for bupropion [101]. The incidence of TESD in patients treated with bupropion is similar to placebo, and in certain cases bupropion can even improve sexual functioning compared to previous levels, improving sexual desire. The combination of bupropion with an SSRI can lessen TESD when switching medication is not possible [102]. A grouped analysis of bupropion data found an association with lesser orgasmic dysfunction, lesser sexual arousal disorder, and lesser sexual desire disorder when compared to an SSRI. In this study, the risk of SD when in treatment with bupropion was found to be the same as placebo [103]. Clinical experience has shown that switching from bupropion to an SSRI can lead to SSRI discontinuation syndrome, and it is therefore recommended to include a washout period before beginning bupropion treatment to avoid confounding the effects of the withdrawal from the previous drug with the possible side effects of bupropion, which can lead to early drop-out.

Despite the proven efficacy of bupropion in treating depression, its side effects [50] may worsen symptoms of anxiety (nervousness, agitation) in anxiety disorders, and clinicians should thus consider other options.

##### Switching to Agomelatine 

Agomelatine stimulates the MT1 and MT2 melatonin receptors while simultaneously antagonizing 5HT2C receptors, thus achieving a noradrenergic and dopaminergic effect and thereby preserving sexual function [34]. The results of a study by Kennedy et al. [33] in Canada comparing agomelatine with venlafaxine showed that patients treated with 50 mg of agomelatine had better sexual function that those receiving 150 mg of venlafaxine XR after 12 weeks of treatment. Both drugs showed comparable antidepressive efficacy. An extension study was performed only on those patients who were free of depression at the end of the previous study and confirmed the improvement of sexual function with agomelatine treatment.

Two double-blind, placebo-controlled studies were performed with healthy male volunteers comparing agomelatine (25–50 mg/day) with paroxetine [19] or escitalopram with the aim of eliminating confounding factors related to SD [20]. More than 80% of subjects from the paroxetine group showed sexual problems in the first week (mainly ejaculation retardation and anorgasmia), while the sexual problems reported by the two agomelatine groups were similar to placebo. These results were replicated in another double-blind, placebo-controlled study with escitalopram [20].

##### Switching to Mirtazapine

Mirtazapine is a noradrenergic and serotoninergic dual-action AD that stimulates 5HT1 postsynaptic receptors and blockades 5HT2 and 5HT3 postsynaptic receptors. Mirtazapine also increases serotoninergic activity by blocking presynaptic α-2 self-receptors found in serotoninergic neurons [104]. The drug has a low incidence of TESD in depressed patients [105] and has been shown to improve previously altered sexual function in depressed patients [106]. Observational studies have shown an improvement in TESD in 70%–90% of patients after switching their existing AD to mirtazapine [107], although weight gain and sedation can appear after such a switch. In another study of five patients, TESD was found to be improved (mainly erectile dysfunction and orgasm problems) by switching their existing AD to mirtazapine [108]; however, in a separate study of four patients, sexual function did not improve when mirtazapine was added to previous AD treatment as an add-on treatment.

In a six-month Spanish group study of 55 patients whose AD was switched to mirtazapine [109], a significant improvement was observed across all five dimensions of the PRSexDQ questionnaire (low libido, orgasm retardation, anorgasmia, erectile dysfunction/vaginal lubrication, acceptance of dysfunction) from the first month of mirtazapine treatment and continued up until the end of the study. While retaining clinical efficacy, tolerability was rated as good or very good in 81.1% of the patients, with weight gain and sedation being observed as adverse effects. Weight gain was badly tolerated by women, and some of them considered returning to the previous treatment despite the sexual dysfunction.

It is necessary to adequately consider all of the drugs mentioned above before switching AD medication, since many women are sensitive to weight gain and have a low acceptance of this side effect.

##### Switching to Other Antidepressants

Switching to moclobemide, a reversible monoamine oxidase type A (MAO-A) inhibitor, has been shown to improve sexual function (in 73.3% of patients) at doses between 450 and 600 mg/day [110]. Generally, the incidence of TESD with moclobemide treatment does not exceed 10%, and in fact the drug can have a stimulating effect on sexual function [111]. Libido decrease and the frequency of ejaculation retardation are similar to placebo, while the percentages with phenelzine (a classic, non-reversible monoamine oxidase inhibitor (MAOI)) were found to be 28.6% and 42.8%, respectively [112]. The use of moclobemide has been limited due to reasonable doubts about its utility in the long term, the hypothetical need to increase dosing, and its disputable efficacy in severe depression.

Reboxetine has been proven to be associated with a lower incidence of TESD in comparative studies with SSRIs and switching to this AD could be an adequate choice to alleviate TESD [54]. However, the possible adrenergic side effects (anxiety, insomnia, xerostomia) should be considered. In theory, orgasm will not be affected by reboxetine use as it is directly related to serotoninergic mechanisms. Clinicians use this drug empirically for this reason; however, no studies have assessed the use of reboxetine to alleviate TESD. However, adrenergic drugs may not be useful in cases where an SSRI is the first choice (obsessive-compulsive disorder or panic disorder).

The results of an Iranian single-blind randomized controlled trial suggest that trazodone (a 5HT1A receptor agonist) could be an optimal alternative to SSRIs [113]. Additionally, a recent literature review [114] suggests that the balance between efficacy and tolerability in trazodone might be better than expected at much lower doses than the available formulations (optimal dose being no more than 19.2 mg daily).

Very few studies have investigated switching AD to maprotiline or mianserin as a management strategy for TESD.

##### Switching to a Partially Serotoninergic Antidepressant

Switching to non-serotoninergic ADs has important limitations, such as the existence of pathologies where SSRIs should be the first option (e.g., obsessive-compulsive disorder, panic attack disorder, impulse control disorder, and eating disorders).

Desvenlafaxine [43], which has a smaller serotoninergic profile than venlafaxine, appears to cause less TESD at 50 mg, however TESD can appear at doses above 100 mg [46,70]. Recent observational data from a switching study in which inclusion and exclusion criteria were controlled showed that switching to desvenlafaxine led to a decrease of severe TESD from 93.3% to 75.6% [56] and an improvement of sexual desire and orgasmic dysfunction, but no improvement in sexual arousal. This study also showed an improvement in the tolerance to TESD; the risk of dropout decreased from 26.7% at baseline to 11.1% in the follow-up visit after six months. Furthermore, in another study with the same methodology using the PRSexDQ questionnaire, desvenlafaxine showed a lower rate of TESD when it was prescribed as the first option (44.4%) compared to venlafaxine (74%) and duloxetine (75%) [18]. The lower pharmacodynamic potency of desvenlafaxine in the reuptake of serotonin compared to venlafaxine might be linked to its lower effect in the dopaminergic brake, mediated by serotonin, which is associated with sexual dysfunction.

Vortioxetine, a multimodal drug with a low serotoninergic effect compared to SSRIs, could be another alternative for alleviating TESD. In two controlled switching studies by Jacobsen [42,115], patients that were being treated with an SSRI were switched to either escitalopram or vortioxetine. The results showed at least similar efficacy for both of the comparisons, however a significant improvement in sexual function was observed in the vortioxetine group. This improvement covered four out of the five dimensions (desire/frequency, desire/interest, arousal/erection, and orgasm) and all three phases (desire, arousal, and orgasm) of sexual functioning. To our knowledge no studies have been conducted on switching to duloxetine due to TESD.

Among all of the cited methods, it is not easy to select the best one for a general recommendation. An individualized selection is recommended which takes each symptom and patient tolerance into account. However, switching to bupropion is supported by the most scientific evidence and clinical experience, while switching to agomelatine is well supported by placebo-controlled studies in healthy non-depressed volunteers.

#### 3.3.6. Non-Pharmacological Measures

Psychoeducation always plays an important role in the treatment of depression and is a necessary tool in chronic TESD that affects the patient’s partner. Preparing patients for the possible presentation of TESD, as well as evaluating their acceptance of TESD if it presents, can greatly help patients to deal with the related uncertainties. 

When performed before sexual relations, physical exercise improves sexual desire and sexual satisfaction in women with ADs [116]. In a controlled trial with depressed males, massages with *Rosa damascena* oil improved TESD caused by SSRIs compared to placebo [117].

Some patients may not wish to change their treatment or may not respond to suggested TESD alleviation strategies. In these cases, it is desirable to build collaborations with licensed sex therapists who can work with patients on issues such as low desire and who may be able to help patients learn to broaden their concept of sexual pleasure and sexual intimacy with a partner.

Surrogate partner therapy has been suggested as a potential means of managing sexual dysfunction [118], however there is minimal data supporting this and to date no controlled trials have been published. This may be due to the fact that surrogate partner therapy is poorly understood by health professionals, as well as the ethical and legal concerns which surround such therapy. The several contraindications of surrogate partner therapy (individuals who appear easily capable of establishing relationships, individuals already in committed partner relationships, individuals who have a history or suspected history of psychiatric instability) seem to make it unsuitable for alleviating TESD.

## 4. Clinical Recommendations

There is unanimous agreement that the incidence of TESD is currently underestimated by clinicians and that clinicians rarely ask patients about their sexual activity (be it due to a lack of time, preparation, or interest). AD treatment, which is very often necessary in the long-term or even indefinitely, can compromise adherence, thus worsening the patient’s quality of life and that of their partner and enabling relapses and recurrences. There is a risk of overstating the role of depression and anxiety disorders in causing TESD. However, given the data, it is essential to gather a full psychosexual history before the start of AD medication. Subsequently, clinicians should monitor possible changes in sexual function, and should select an individualized strategy for the management of TESD if it appears and is badly tolerated by the patient or their partner. There is not a single or standardized conceptualization of human sexuality, and approaches to alleviate TESD therefore remain dependent on clinicians’ professional interests, which almost always leave the patient’s assessment incomplete.

The main focus of clinicians should be on primary prevention, and starting a treatment with drugs that preserve sexual function should always be the first choice in sexually active patients who require long-term treatment.

Agomelatine, mirtazapine, and bupropion are some of the better ADs to alleviate TESD. However, in situations where it is important to retain some serotonergic efficiency, desvenlafaxine and vortioxetine at lower doses seem to be the best options. If prevention is not possible, there are several strategies to manage TESD symptomatology. The literature data reviewed in the present study (albeit scarce), as well as recommendations based on the level of scientific evidence [119,120], allow us to recommend different intervention strategies according to the key TESD symptoms following the Scottish Intercollegiate Guidelines Network Grading Review Group (Table 4). 

In any case, clinicians should tailor each strategy to the individual patient, addressing its impact on the patient’s TESD and its possible improvements and changing to alternative treatments if the original treatment is not effective. The improvement of TESD will lead to a better tolerability of antidepressant treatment, better treatment adherence, better patient quality of life, and the best available global outcomes.

## 5. Conclusions

TESD possible solutions should primarily be focused on prescribing an antidepressant with a low risk for SD. Agomelatine, mirtazapine, and bupropion are some of the better options, though where the serotonergic effect is needed, desvenlafaxine and vortioxetine at lower doses work adequately. If prevention is not possible, a number of other strategies to manage TESD symptomatology should be considered, such as adding another drug or switching to a different antidepressant. These strategies should always be carefully chosen with regard to the patient’s individual situation and symptomatology, weighing down the benefits against the costs.

## Figures and Tables

**Table 1 jcm-08-01640-t001:** Meta-analysis of the prevalence of sexual dysfunction in patients taking antidepressants [3].

Antidepressant	Prevalence of Sexual Dysfunction	Main Form of Sexual Dysfunction
Moclobemide	0.22%	Desire (4.11%), orgasm (0.41%), arousal (1.91%)
Agomelatine	0.25%	Desire (1.52%), orgasm (1.31%)
Amineptine	0.46%	Insufficient data
Nefazodone	0.46%	Desire (1.53%), orgasm (0.32%), arousal (0.19%)
Bupropion	0.75%	Desire (1.29%), orgasm (1.26%), arousal (1.83%)
Mirtazapine	2.32%	Desire (6.03%), orgasm (4.4%), arousal (3.92%)
Fluvoxamine	3.27%	Desire (6.31%), orgasm (11.91%), arousal (31.42%)
Escitalopram	3.44%	Desire (1.10%), orgasm (4.23%), arousal (0.68%)
Duloxetine	4.36%	Desire (5.25%), arousal (10.95%)
Phenelzine	6.43%	Desire (5.71%), orgasm (11.85%), arousal (5.76%)
Imipramine	7.24%	Desire (6.33%), orgasm (5.25%), arousal (6.07%)
Fluoxetine	15.59%	Desire (45.59%), orgasm (11.91%), arousal (31.42%)
Paroxetine	16.68%	Desire (46.99%), orgasm (18.45%), arousal (44.44%)
Citalopram	20.27%	Desire (55.30%), orgasm (14.39%), arousal (82.48%)
Venlafaxine	24.82%	Desire (23%), orgasm (15.94%), arousal (54.04%)
Sertraline	27.43%	Desire (42.95%), orgasm (15.03%), arousal (38.58%)

**Table 2 jcm-08-01640-t002:** Rates of spontaneous remission of treatment emergent sexual dysfunction (TESD) [2,16,71,72,73].

Study	Time Elapsed	Partial Recovery	Full Recovery	Total
Nurnberg and Levine [71]	>3 and <6 months (*n* = 2)	0	2	2
	<3 months (*n* = 1)	0	1	1
Montejo et al. [2]	6 months (*n* = 156)	20 (12.82%)	9 (5.8%)	29 (18.59%)
Ashton and Rosen [69]	>6 months (*n* = 132)	13 (9.8%) ^1^		13 (9.8%)
Montejo et al. [16]	>6 months (*n* = 143)	14 (11.2%)	16 (9.7%)	30 (20.97%)
	>3 and <6 months (*n* = 131)	10 (7.6%)	5 (3.8%)	15 (11.4%)
	<3 months (*n* = 78)	9 (11.5%)	1 (1.7%)	10 (12.8%)

^1^ The authors did not differentiate between partial and full recovery.

**Table 3 jcm-08-01640-t003:** Add-on treatments for antidepressant-induced anorgasmia [77,78,79,80,81,82,83,84].

Drug	Mechanism of Action	Dose (mg/day)
Ciproheptadine	5HT antagonism	4–8
Buspirone	5HT1A partial agonism	14–45
Yohimbine	α-2 adrenergic antagonism	5–10
Amantadine	Dopaminergic agonist	100–400
Metilphenidate	Dopaminergic agonist	10–30
Bupropion	Dopaminergic and adrenergic effect	150–300
Mirtazapine	5HT2 antagonism	15–45

**Table 4 jcm-08-01640-t004:** Clinical recommendations for alleviating TESD based on scientific evidence levels * [42,50,56,96,117,118].

Symptom	Alternative 1	Evidence Level	Alternative 2	Evidence Level
**Low sexual desire**	Switching to agomelatine.	A	Switching to desvenlafaxine (50 mg/day) or vortioxetine (<15 mg/day).	B
	Switching to a non-serotoninergic drug (bupropion or mirtazapine).	B	Dose reduction; associating aripiprazole.	C
	Adding bupropion.	B		
**Orgasm retardation**	Switching to agomelatine.	A	Switching to desvenlafaxine (50 mg/day) or vortioxetine (<15 mg/day).	B
	Switching to a non-serotoninergic drug or fluvoxamine.	B	Dose reduction.	C
**Anorgasmia**	Switching to agomelatine.	A	Switching to desvenlafaxine (50 mg/day) or vortioxetine (<15 mg/day).	B
	Switching to a non-serotoninergic drug or fluvoxamine.	B	Dose reduction or “weekend holiday” protocol.	C
**Erectile dysfunction**	Switching to agomelatine.	A	Switching to desvenlafaxine (50 mg/day) or vortioxetine (<15 mg/day).	B
	Switching to a non-serotoninergic drug.	B	Associate PD-5 inhibitors.	
**Scarce vaginal lubrication**	Switching to agomelatine.	A	Switching to desvenlafaxine (50 mg/day) or vortioxetine (<15 mg/day).	B
	Switching to a non-serotoninergic drug.	B	Dose reduction; using vaginal lubricants.	C

* A: Recommended (good evidence that the measure is effective, and the benefits far outweigh the harms). B: Recommended (at least moderate evidence that the measure is effective, and the benefits outweigh the harms). C: Neither recommended nor inadvisable (at least moderate evidence that the measure is effective; however, the level of benefit is very similar to the level of harm and a general recommendation cannot be justified).

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
