# Peer review of "Management Strategies for Antidepressant-Related Sexual Dysfunction: A Clinical Approach"

_jcm, 2019, doi:10.3390/jcm8101640_

Round 1

Reviewer 1 Report

I appreciate the addition of description of methodology, but found the description a bit unclear.  For example: "Articles focusing on TESD as a phenomenon but devoid of management strategies were carefully selected to be representative, with 32 such studies being included in the introduction section of the present study."  It is not clear how these 32 were chosen from the 124 originally identified, and I don't know what the 'introduction section' of this paper refers to.  I would also not characterize this review as a study, but that is a minor point.

The additional data referenced and detail added has strengthened this paper; which provides a useful overview of treatment of TESD.

Author Response

Author’s  reply. Reviewer 1. Round 2. jcm-577059

Dear reviewer.

We thank you very much for the time spent reviewing this article and the kind suggestions for improvement. Below you will find the answers to your comments and suggestions point by point.

Comment 1. I appreciate the addition of description of methodology, but found the description a bit unclear.  For example: "Articles focusing on TESD as a phenomenon but devoid of management strategies were carefully selected to be representative, with 32 such studies being included in the introduction section of the present study."  It is not clear how these 32 were chosen from the 124 originally identified, and I don't know what the 'introduction section' of this paper refers to.  I would also not characterize this review as a study, but that is a minor point.

Answer: The mentioned Methodology section has been rewritten, in order to be as clear as possible, as follows:

(…) We excluded articles which focused on SD as a general entity rather than a consequence of treatment with ADs.

Initially, 124 relevant results were found which were focused on TESD. Upon manual screening and categorizing, these results included 1) those which acknowledged the prevalence and clinical relevance of the phenomenon, and 2) those focused on clinical management strategies for TESD.

The results described under “1)” were mainly used for the “Introduction” section of the paper. They were a total of 32 articles, including 4 transversal and cross-sectional studies, 11 observational studies, and a total of 17 reviews (of which 5 were meta-analyses). Several of these articles were also used on other sections of the paper to provide background, since many of them are antidepressant-specific.

The results described under “2)” were included in the “Results and clinical approach” section of the paper. They were a total of 72 articles, including 29 reviews (of which seven were systematic reviews or meta-analyses), 19 experimental studies and 23 observational studies. An additional twelve articles were included after selected areas of the paper were revised (experimental studies with sildenafil, trazodone studies, sex surrogate studies, and validated questionnaires about SD).

Reviewer 2 Report

This is a well-written review regarding management strategies for antidepressant-related sexual dysfunction. This is a clinical work, although it seems to me that it would be worth mentioning several experimental works closely related to the subject of this review paper. Although the results obtained in the experimental studies cannot be directly transferred to clinical practice, it is worth mentioning some experimental studies performed in mice. Recently, PDE5 inhibitor sildenafil has been shown to potentiate antidepressant-like activity of sertraline, maprotiline, trazodone (PMID: 28013355), bupropion, venlafaxine (PMID: 22940586), mianserin (PMID: 22406168) and amitriptyline (PMID: 2221520). On the other hand, the anti-immobility action of paroxetine was reduced (PMID: 23238482).

Author Response

Author’s  reply. Reviewer 2. Round 2. jcm-577059

Dear reviewer.

We thank you very much for the time spent reviewing this article and the kind suggestions for improvement. Below you will find the answers to your comments and suggestions point by point.

Comment 1. This is a well-written review regarding management strategies for antidepressant-related sexual dysfunction. This is a clinical work, although it seems to me that it would be worth mentioning several experimental works closely related to the subject of this review paper. Although the results obtained in the experimental studies cannot be directly transferred to clinical practice, it is worth mentioning some experimental studies performed in mice. Recently, PDE5 inhibitor sildenafil has been shown to potentiate antidepressant-like activity of sertraline, maprotiline, trazodone (PMID: 28013355), bupropion, venlafaxine (PMID: 22940586), mianserin (PMID: 22406168) and amitriptyline (PMID: 2221520). On the other hand, the anti-immobility action of paroxetine was reduced (PMID: 23238482).

Answer: An additional paragraph has been written on Section 3.3.3 (Potentiation Strategy) regarding PD-5 inhibitors and Sildenafil that includes the references provided:

(…) Additionally, these inhibitors may have an antidepressive effect by blocking central cholinergic receptors; recent experimental studies with mice have proved that administration of sildenafil potentiates antidepressant-like activity of sertraline, maprotiline, trazodone [89], bupropion, venlafaxine [90], mianserin, tianeptine [91], and amytriptiline [92]. However, sildenafil’s mechanism of action may be directly opposed to the one used by paroxetine [93], and when used together the antidepressant effect may prove diminished or nullified.